The Iflaviruses Sacbrood virus and Deformed wing virus evoke different transcriptional responses in the honeybee which may facilitate their horizontal or vertical transmission

Ryabov Eugene V. 1 eugene.ryabov@warwick.ac.uk eugene.ryabov@gmail.com
Fannon Jessica M. 1
Moore Jonathan D. 2
Wood Graham R. 2
Evans David J. 3
1 School of Life Sciences, University of Warwick , Coventry , United Kingdom
2 Warwick Systems Biology Centre, University of Warwick , Coventry , United Kingdom
3 Biomedical Sciences Research Complex, University of St. Andrews , St. Andrews , United Kingdom
Newton Irene
Electronic publication date: 2016 Jan 18
Publication date: 2016
Volume: 4
Electronic Location ID: e1591
Received 2015 Sep 29; Accepted 2015 Dec 21
Copyright: ©2016 Ryabov et al.
Copyright year: 2016
Copyright holder: Ryabov et al.
License: This is an open access article distributed under the terms of the Creative Commons Attribution License, which permits unrestricted use, distribution, reproduction and adaptation in any medium and for any purpose provided that it is properly attributed. For attribution, the original author(s), title, publication source (PeerJ) and either DOI or URL of the article must be cited.
License URL: https://creativecommons.org/licenses/by/4.0/

Keywords: Antimicrobial peptide, Honeybee, RNA virus, Innate immunity, Apis mellifera, RNA-Seq, Iflavirus, Transcriptome, Sacbrood virus, Deformed wing virus

Funding: BBSRC BB/M00337X/1 Insect Pollinators Initiative BBI0008281 This work was supported by the Biotechnology and Biological Sciences Research Council (BBSRC), the Department for Environment, Food and Rural Affairs, the Natural Environment Research Council, the Scottish Government and the Wellcome Trust, under the Insect Pollinators Initiative, grant number BBI0008281, and BBSRC grant BB/M00337X/1. The funders had no role in study design, data collection and analysis, decision to publish, or preparation of the manuscript.

==============================
Sacbrood virus (SBV) and Deformed wing virus (DWV) are evolutionarily related positive-strand RNA viruses, members of the Iflavirus group. They both infect the honeybee Apis mellifera but have strikingly different levels of virulence when transmitted orally. Honeybee larvae orally infected with SBV usually accumulate high levels of the virus, which halts larval development and causes insect death. In contrast, oral DWV infection at the larval stage usually causes asymptomatic infection with low levels of the virus, although high doses of ingested DWV could lead to DWV replicating to high levels. We investigated effects of DWV and SBV infection on the transcriptome of honeybee larvae and pupae using global RNA-Seq and real-time PCR analysis. This showed that high levels of SBV replication resulted in down-regulation of the genes involved in cuticle and muscle development, together with changes in expression of putative immune-related genes. In particular, honeybee larvae with high levels of SBV replication, with and without high levels of DWV replication, showed concerted up-regulated expression of antimicrobial peptides (AMPs), and down-regulated expression of the prophenoloxidase activating enzyme (PPAE) together with up-regulation of the expression of a putative serpin, which could lead to the suppression of the melanisation pathway. The effects of high SBV levels on expression of these immune genes were unlikely to be a consequence of SBV-induced developmental changes, because similar effects were observed in honeybee pupae infected by injection. In the orally infected larvae with high levels of DWV replication alone we observed no changes of AMPs or of gene expression in the melanisation pathway. In the injected pupae, high levels of DWV alone did not alter expression of the tested melanisation pathway genes, but resulted in up-regulation of the AMPs, which could be attributed to the effect of DWV on the regulation of AMP expression in response to wounding. We propose that the difference in expression of the honeybee immune genes induced by SBV and DWV may be an evolutionary adaptation to the different predominant transmission routes used by these viruses.

Introduction

The Western honeybee, Apis mellifera, is the most important managed insect pollinator worldwide. In recent decades a global decline in the number of honeybee colonies was reported, threatening security of the global food supply (Vanbergen et al., 2013), with pathogens—in particular viruses—contributing significantly to these declines. These viral pathogens are predominantly single-stranded, positive sense RNA viruses of the families Dicistroviridae and Iflaviridae, and may exhibit differing virulence levels, causing infections ranging from asymptomatic to acute and resulting in rapid insect death (McMenamin & Genersch, 2015). It is also apparent that some viruses exhibit strain differences in virulence. For example, the most widespread honeybee virus, Deformed wing virus (DWV) (De Miranda & Genersch, 2010; Lanzi et al., 2006) and the very closely related variants Varroa destructor virus-1 (Ongus et al., 2004) and Kakugo virus (Fujiyuki et al., 2004), usually cause asymptomatic infections with low levels of the virus when transmitted vertically or orally. In contrast, DWV transmission by the ectoparasitic mite Varroa destructor—by direct injection to the honeybee haemolymph—results in the selection of highly pathogenic strains of DWV with significantly reduced genetic diversity (Martin et al., 2012; Ryabov et al., 2014) which accumulate to very high levels in infected pupae and cause characteristic symptoms, including deformed wings and shortened abdomen. The doses of DWV, including its virulent strains, which are delivered orally to larvae during brood rearing cause only asymptomatic infections and accumulate to low levels, making it possible for infected honeybees to survive to adulthood and transmit the virus horizontally or vertically (Ryabov et al., 2014; Yue & Genersch, 2005). In contrast to DWV, Sacbrood virus, (SBV, a related member of the Iflaviridae) a predominant viral pathogen in the Asian honeybee Apis cerana, accumulates to high levels and causes acute infections in orally inoculated honeybee larvae (Ai, Yan & Han, 2012; Bailey, Gibbs & Woods, 1964; Ghosh et al., 1999). SBV infection has a much more pronounced impact on honeybee development than DWV; honeybee larvae with high levels of the virus have a gondola-shaped sac-like appearance with tough leathery skin and die before pupation. It is likely that SBV is transmitted from the larvae killed by SBV to in-hive worker honeybees, which subsequently transmit the virus to young larvae (Bailey, 1969).

In this study, we analyzed global honeybee transcriptional responses to both DWV and SBV using RNA-Seq. We further analyzed the impact of DWV and SBV on the expression of several immune related genes of the honeybee by real-time PCR (qRT-PCR). We found that different sets of genes were differentially expressed (DE) in honeybee larvae with high levels of either DWV alone or SBV and DWV combined, and that high levels of SBV infection had a more significant impact on global gene expression in the honeybee compared to high levels of DWV, in particular on the expression of immune-related genes. We found, in both larval feeding and pupal injection experiments, that high levels of SBV were associated with up-regulation of the expression of antimicrobial peptide (AMP) genes and changes in expression of the genes involved in regulation of melanisation, which may suppress this function. These differential effects of DWV and SBV on expression of AMP and melanisation pathway genes may be an adaptation of these viruses to facilitate their vertical and horizontal routes of transmission respectively.

Materials & Methods

Honeybee rearing, virus preparations and inoculation

Colonies of healthy Western honeybees (Apis mellifera) with low managed levels of Varroa destructor infestation were maintained in Warwickshire, UK, and used as a source of larvae and pupae. DWV virus preparation was isolated from honeybee pupae sourced from a colony with high Varroa infestation levels. Virus preparations containing both SBV and DWV (SBV + DWV) were purified from larvae and pupae of the Varroa-infested colonies where some larvae showed typical SBV-induced symptoms. No other known honeybee viruses were detected in the prepared virus stocks. Virus isolation was carried out as described previously (Moore et al., 2011) and the virus preparations were stored at −80 °C prior to use. For inactivation, the virus preparations were irradiated by UV light (Simonet & Gantzer, 2006).

Artificial rearing of the honeybee larvae was carried out essentially as described previously (Aronstein & Saldivar, 2005; Vandenberg & Shimanuki, 1987). For oral inoculation newly hatched honeybee worker larvae were transferred to an artificial honeybee larval diet and maintained at +33 °C. After 12 h the larvae were orally inoculated with a single dose of the virus preparation containing SBV and DWV. Approximately 1010 SBV and 1010 DWV virus particles (SBV + DWV) were added to 50 µl of the honeybee rearing diet per bee, which was consumed within 12 h. No virus was added to the subsequent portions of the larval food. The controls in the feeding experiment included virus-free phosphate-buffered saline (PBS), and the UV-inactivated SBV + DWV virus preparation (UV-inactivated virus, SBV + DWV). The larvae were maintained for an additional 9 days up to the late fourth instar stage. Whole-body RNA samples were extracted from individual insects at 4 days post inoculation (dpi) or 9 dpi.

For the honeybee pupa infection, worker pupae sourced at the white eye stage (12th–13th days of development) received injections into the haemolymph using a syringe with a 0.3 mm outer diameter needle (Ryabov et al., 2014) either with 10 µl of phosphate-buffered saline (PBS), or with DWV preparations (106 DWV virus particles in PBS), or with the mixture of SBV and DWV (106 SBV and 106 DWV virus particles in PBS). The pupae were reared at +33 °C as previously described (Ryabov et al., 2014). Whole-body RNA samples were extracted from individual pupae at 2 dpi or 5 dpi.

Gene expression analysis

Total RNA was extracted from individual pupae with Tri-reagent (Trizol) (Ambion, Foster City, CA, USA) according to the manufacturer’s instructions. The extracted column-purified total RNA from individual honeybees was used for high-throughput sequencing of the mRNA populations by RNA-Seq. The experiment and the reads were deposited into the European Nucleotide Archive under accession number PRJEB6511.

Quantification of viral RNA and the honeybee transcripts were carried out by quantitative reverse transcription PCR (qRT-PCR) as described previously (Ryabov et al., 2014). In brief, the RNA samples were treated with RNA-free DNAse1 (New England BioLabs), purified using RNAeasy plant mini kit (Qiagen, Hilden, Germany) and used for cDNA synthesis with random hexanucleotide primers. qRT-PCR reactions were performed using SYBR Green kit (Ambion, Foster City, CA, USA) with the primers to viral RNA and to the honeybee transcripts (Table S1). Tukey’s Honest Significant Difference test (Tukey’s HSD) was used to determine significantly different virus and gene expression levels (Tukey, 1949).

Bioinformatics

The RNA-Seq reads were aligned using Bowtie2 (Langmead et al., 2009) (with the least stringent alignment settings to allow detection of the sequence variants, “–very-sensitive” option) to the latest honeybee transcriptome annotation (OGS3; containing 16,041 putative transcripts), as well as to a set of sequences of the known fungal and viral pathogens of the honeybees used previously (Bull et al., 2012; Ryabov et al., 2014). We used samtools idxstats to produce a summary of the number of reads aligning to the honeybee transcriptome and the DWV and SBV reference sequences (GenBank accession numbers NC˙004830and AF092924respectively). The Next Generation Sequencing (NGS) gene expression profiles were used to identify differentially expressed (DE) genes using DESeq (Anders & Huber, 2010) and edgeR (Robinson, McCarthy & Smyth, 2010), with adjusted p-values and a false discovery rate (FDR) below 0.05. Drosophila homologues of the honeybee genes were identified previously (Ryabov et al., 2014) and those DE in the contrasts were used for Gene Ontology (GO) analysis (Ashburner et al., 2000) using AmiGO (Carbon et al., 2009).

Results

Oral infection of honeybee larvae with DWV and SBV

Artificially reared honeybee worker larvae were orally inoculated with virus preparations containing SBV and DWV (“SBV + DWV”) and controls included UV-inactivated “SBV + DWV” virus preparation and PBS (Fig. 1A). The doses of both DWV and SBV, 1010 genome equivalents, were sufficient to allow replication of the viruses to high levels when ingested at the larval stage (EV Ryabov and DJ Evans, pers. comm., 2015). Notably, this DWV dosage was 100 times higher than a dose used in oral infection of the adult bees which did not result in establishing high levels of DWV infection (Moeckel, Gisder & Genersch, 2011). Quantification of SBV and DWV in the experimental insects assayed at 9 dpi showed that individuals of both control groups had low levels of both DWV and SBV (Ct values 31–22, and 32–24 respectively), while among the virus-fed insects there were individuals with high levels of either DWV or SBV, as well as those with high levels of both viruses (Ct values 8–14, and 9–15).

For comprehensive characterization of honeybee gene expression in response to high levels of DWV and SBV, we used an RNA-Seq approach. The analysis was carried out using whole-body RNA extracted from individual honeybee pupae sampled at 9 dpi. Controls included pupae with low levels of DWV and SBV (samples 1 and 2) for comparison with the three virus-infected samples; one of these (sample 3) had high level of DWV and low level of SBV, and two samples had high levels of both SBV and DWV (samples 4 and 5) (Fig. 1A and Table 1).

Figure 1 Schematic representation of experimental infection of honeybees with SBV and DWV, (A) larval oral inoculation, and (B) pupal haemolymph injection.

Approximately 10 million 101 nt reads were produced for each library (Table 1) and were aligned to the latest honeybee transcriptome annotation (OGS3) and to the sequences of known fungal and viral pathogens of the honeybees used previously (Bull et al., 2012; Ryabov et al., 2014). Apart from DWV-like viruses and SBV (GenBank accession numbers NC˙004830and AF092924respectively) no other pathogens were detected. We observed a dramatic increase of DWV and SBV coverage, normalized to host actin mRNA coverage (GB44311), in infected honeybees compared to controls (Table 1). For example, there was an ∾1,000-fold increase in DWV reads in virus-infected pupae (samples 3, 4 and 5) compared to controls (samples 1 and 2), from 0.05 to ∾50 in concordance with previously reported actin-normalized levels of DWV in pupae with low (0.1 genomes/actin mRNA) and high (10–100 genomes/actin mRNA) levels of DWV by qRT-PCR (Moore et al., 2011; Ryabov et al., 2014). SBV levels showed over 1,000-fold increase in samples 4 and 5 compared to the control samples (1 and 2) and sample 3; the ratios of SBV to actin read coverage increased from 0.04–0.20 to 378–573 (Table 1). The observed increase of the SBV load was similar to previously reported differences between SBV levels in asymptomatic honeybee larvae with low SBV levels and the symptomatic larvae with high SBV (Blanchard et al., 2014).

Table 1 Summary of the NGS libraries of the larval oral inoculation experiment.

Sample ID	Treatment group	ENA sample accession	Total reads	A. mellifera OGS3, mRNA reads	Total DWV reads (Aligned to GenBank accession number NC˙004830)	DWV to actin mRNA (GB44311) coverage ratio	Total SBV reads (Aligned to GenBank accession number AF092924)	SBV to actin mRNA (GB44311) coverage ratio	
1	Control	SAMEA2591288	9,691,343	6,842,703	7,555	0.047	28,541	0.201	
2	Control	SAMEA2591289	10,630,145	6,204,592	6,210	0.049	4,009	0.036	
3	DWV	SAMEA2591290	9,785,423	3,352,681	3,240,468	55.263	6,179	0.120	
4	BV + DWV	SAMEA2591291	10,069,125	645,114	736,640	34.841	7,021,099	378.684	
5	BV + DWV	SAMEA2591292	10,257,560	604,367	887,349	54.627	8,171,254	573.641	

RNA-Seq analysis reveals that high levels of DWV, and SBV with DWV co-infection, evoke different transcriptional responses in orally infected honeybee larvae

We stratified the RNA-Seq samples according to the levels of DWV and SBV (high and low) into three groups, “Control” (samples 1 and 2), “DWV” (sample 3), and “SBV + DWV” (samples 4 and 5) and, by using both DESeq (Anders & Huber, 2010) and edgeR (Robinson, McCarthy & Smyth, 2010), identified differentially expressed (DE) genes in five contrasts (Fig. 2, Table S2) to assess the effect of virus infections on the host gene expression. Potential functional consequences of DE were inferred following overrepresented Gene Ontology (GO) analysis (Ashburner et al., 2000; Table S3).

Figure 2 Effect of virus infection on global honeybee gene expression, RNA-Seq experiment: experimental groups and contrasts.

Arrows indicate direction of the contrasts (head against tail). The numbers of differentially expressed (DE) honeybee genes and of DE immune-related genes are shown for each of the contrasts.

The highest numbers of DE genes were identified in the contrasts involving the “SBV + DWV” group. Of these, contrast 4 (high SBV + DWV vs control) had 1,638 DE genes, which included almost all (1,076 of 1,088) of those identified as DE in Contrast 2 (high SBV + DWV vs. control and high DWV alone). High commonality, 697 of 824 genes, was also observed between the DE genes in Contrast 5 and Contrast 4 (high SBV + DWV vs. high DWV alone and high SBV + DWV vs. control respectively (Fig. 2). The direction of gene expression change was the same (e.g., genes up-regulated in Contrast 4 were also up-regulated in Contrast 5).

The number of DE genes in Contrast 3 (transcriptome changes associated with high DWV levels alone) was lower than those observed in Contrasts 2, 4 and 5, all of which involved the SBV + DWV group (Fig. 2). A very low number of genes (n = 4) was identified in Contrast 1 (common response to high levels of DWV alone and high levels of SBV + DWV), and the low commonality between Contrasts 3 and 4 (n = 9) strongly suggested that transcriptional responses to high levels of DWV and SBV + DWV were different (Fig. 2). Indeed, GO analysis (Table 2, Table S3) showed that different overrepresented GO terms were associated with the DE genes in Contrast 3 (high DWV levels) compared with the genes in Contrasts 2, 4, and 5 (high levels of SBV and DWV), providing further evidence that high level replication of SBV or DWV affected different biological processes in the honeybee. When compared with the low virus level control, the insects with high DWV levels (Contrast 3) showed up-regulation of the genes involved in translation, metabolic processes, and ATP metabolism (Table 2, Table S3). Changes in honeybee gene expression associated with high levels of SBV + DWV were more pronounced when compared to those associated with high DWV levels alone. The down-regulated DE genes associated with increased levels of SBV (Contrasts 2, 4, and 5) were involved in cuticle and muscle development (Table 2, Table S3), consistent with the reported phenotypic effects of SBV infection, which include halted development and abnormal cuticle (Bailey, Gibbs & Woods, 1964). Surprisingly, despite very low commonality between Contrasts 3 and 4, a considerable proportion of DE genes in Contrast 3 (68 of 223) were also DE in Contrast 5 (Fig. 2). However, the vast majority of these (67/68) exhibited virus-dependent DE in opposing directions, i.e., genes up-regulated in response to high levels of DWV alone were down-regulated in response to high levels of SBV, even in the presence of high levels of DWV (Table S4). The over-represented GO terms associated with these genes indicated that high levels of DWV induced increased expression of the genes involved in ATP metabolism, whereas high levels of SBV had the opposite effect on the expression of these genes, overriding the effect of DWV on their expression (Table 2, Table S4). In respect to the genes up-regulated in response to high levels of SBV, we were particularly intrigued with the over-representation of GO terms associated with immune response, e.g., “Immune system process,” “Defense response” (Table 2, Table S3).

Table 2 Gene ontology (GO) Biological Process (BP) terms associated with the upregulated and downregulated differentially expressed genes in the honeybees of the larval feeding NGS experiments (only the top 10 over-represented GO PB terms with the lowest p-values are shown).

GO term	P-value	Sample frequency	Background frequency	
Contrast 3	
Upregulated DE genes				
GO:0006412 translation	3.31E–05	31/186 (16.7%)	787/14,580 (5.4%)	
GO:0044237 cellular metabolic process	1.87E–04	96/186 (51.6%)	4,811/14,580 (33.0%)	
GO:0009161 ribonucleoside monophosphate metabolic process	2.16E–04	12/186 (6.5%)	132/14,580 (0.9%)	
GO:0009123 nucleoside monophosphate metabolic process	2.35E–04	12/186 (6.5%)	133/1,580 (0.9%)	
GO:0046034 ATP metabolic process	3.67E–04	11/186 (5.9%)	113/14,580 (0.8%)	
GO:0032543 mitochondrial translation	6.05E–04	6/186 (3.2%)	23/14,580 (0.2%)	
GO:0009167 purine ribonucleoside monophosphate metabolic process	1.11E–03	11/186 (5.9%)	126/14,580 (0.9%)	
GO:0009126 purine nucleoside monophosphate metabolic process	1.11E–03	11/186 (5.9%)	126/14,580 (0.9%)	
GO:0010467 gene expression	4.08E–03	56/186 (30.1%)	2,397/14,580 (16.4%)	
GO:0044249 cellular biosynthetic process	5.38E–03	56/186 (30.1%)	2,418/14,580 (16.6%)	
Downregulated DE genes				
None				
Commonality between Contrasts 2, 4, and 5	
Upregulated DE genes				
GO:0050896 response to stimulus	1.11E–10	104/263 (39.5%)	2,855/14,580 (19.6%)	
GO:0006950 response to stress	3.34E–10	57/263 (21.7%)	1,084/14,580 (7.4%)	
GO:0002376 immune system process	1.04E–09	33/263 (12.5%)	405/14,580 (2.8%)	
GO:0006952 defense response	9.74E–08	29/263 (11.0%)	373/14,580 (2.6%)	
GO:0044699 single-organism process	6.35E–07	194/263 (73.8%)	8,031/14,580 (55.1%)	
GO:0006955 immune response	2.61E–06	24/263 (9.1%)	298/14,580 (2.0%)	
GO:0065007 biological regulation	5.27E–06	109/263 (41.4%)	3,621/14,580 (24.8%)	
GO:0044763 single-organism cellular process	7.96E–06	154/263 (58.6%)	5,930/14,580 (40.7%)	
GO:0045087 innate immune response	6.92E–05	16/263 (6.1%)	157/14,580 (1.1%)	
GO:0044707 single-multicellular organism process	2.72E–04	115/263 (43.7%)	4,170/14,580 (28.6%)	
Downregulated DE genes				
GO:0042335 cuticle development	3.78E-12	27/242 (11.2%)	234/14,580 (1.6%)	
GO:0040003 chitin-based cuticle development	7.15E–11	23/242 (9.5%)	182/14,580 (1.2%)	
GO:0030239 myofibril assembly	8.95E–10	12/242 (5.0%)	37/14,580 (0.3%)	
GO:0055002 striated muscle cell development	6.24E–08	12/242 (5.0%)	51/14,580 (0.3%)	
GO:0055001 muscle cell development	6.24E–08	12/242 (5.0%)	51/14,580 (0.3%)	
GO:0031032 actomyosin structure organization	2.26E–07	13/242 (5.4%)	70/14,580 (0.5%)	
GO:0006030 chitin metabolic process	3.65E–07	16/242 (6.6%)	122/14,580 (0.8%)	
GO:1901071 glucosamine-containing compound metabolic process	1.07E–06	16/242 (6.6%)	131/145,80 (0.9%)	
GO:0006040 amino sugar metabolic process	1.21E–06	16/242 (6.6%)	132/14,580 (0.9%)	
GO:0006022 aminoglycan metabolic process	5.98E–06	16/242 (6.6%)	147/14,580 (1.0%)	

Table 3 Differential expression (DE) of the putative honeybee antimicrobial peptides (AMPs), melanisation, Toll, and Imd pathway genes in the larval feeding experiment.

Fold change values (log2 transformed) are shown only for the genes DE in the contrast. Expression of the genes marked with ⋆ was quantified by qRT-PCR. DE genes were identified by both DESeq and edgeR analyses, with adjusted p < 0.05 and false discovery rate, FDR < 0.05 respectively.

Honeybee gene (OGS3 ID)	Drosophila ortholog (Flybase ID)	Gene name/ description	Pathway, group	Fold change (log2 transformed)	
				Contrast 2	Contrast 3	Contrast 4	Contrast 5	
				High SBV + DWV vs. high DWV and control	High DWV vs. control	High SBV + DWV vs. control	High SBV + DWV vs. high DWV	
GB41428⋆	FBgn0010385	Defensin-1	AMP	9.203	.	9.328	8.738	
GB47318	FBgn0032835	Abaecin	AMP	6.474	.	6.143	10.455	
GB47546		Apidaecin	AMP	5.322	.	5.423	4.925	
GB47618	FBgn0010385	Defensin-2	AMP	10.171	.	9.828	10.730	
GB51223⋆	FBgn0014002	Hymenoptaecin	AMP	7.894	.	8.057	7.410	
GB53576	FBgn0261922	Apisimin	AMP	.	.	2.738	.	
GB50013⋆	FBgn0036891	Prophenoloxidase-activating enzyme (PPAE)	Melanisation	−2.612	.	−2.791	.	
GB48820⋆	FBgn0028985	Serpin (NEC LIKE)	Toll/Melanisation	4.681	.	4.867	4.151	
GB54611	FBgn0028984	Serpin (NEC LIKE)	Toll/Melanisation	2.092	.	2.027	2.359	
GB40699	FBgn0029114	Tollo (Receptor)	Toll	.	.	−1.187	.	
GB43456	FBgn0034476	Toll-7 (Receptor)	Toll	−1.681	.	−1.780	.	
GB49441	FBgn0003450	persephone-Serine protease	Toll	4.182	.	4.134	4.365	
GB54611	FBgn0028984	NEC-like	Toll	2.092	.	2.027	2.359	
GB55007	FBgn0030051	persephone-Serine Protease Immune Response Integrator	Toll	2.067	.	1.975	.	
GB44055	FBgn0000250	cactus (NF-kappa-B inhibitor)	Toll	.	.	2.372	2.457	
GB50418	FBgn0262473	Toll-1 (Receptor)	Toll	2.073	.	2.104	1.962	
GB51741	FBgn0030310	Peptidoglycan recognition protein SA	Toll	2.070	.	2.056	2.119	
GB52631	FBgn0003495	spatzle	Toll	3.224	.	3.284	3.012	
GB51498	FBgn0033402	Myd88	Toll	.	1.549	nd	.	
GB48707	FBgn0024222	immune response deficient 5	Toll	.	1.340	nd	.	
GB42500	FBgn0035976	PGRP-LC	Imd	1.515	.	1.462	1.723	
GB45648	FBgn0013983	imd	Imd	.	.	1.240	.	

Differing effects of DWV and SBV on the expression of immune-related genes

Of 381 putative immune-related genes of the honeybee identified in previous studies (Evans et al., 2006; Ryabov et al., 2014), 98 were DE among the contrasts of the RNA-Seq experiment (Fig. 2, Table S5) with 74 of these genes in contrast 2 (high SBV + DWV vs. high DWV alone and control), 94 of these DE in contrast 4 (high SBV + DWV vs. control), 57 of these genes in contrast 5 (high SBV + DWV vs. high DWV alone) (Fig. 2, Table S5). Strikingly, there were 54 DE immune-related genes shared in contrasts 2, 4 and 5, converging at the high SBV + DWV group (Fig. 2, Table S3). In particular, we observed dramatic up-regulation (30- to 1000-fold) of six antimicrobial peptide (AMP) genes (Table 3, Table S5). Expression of AMPs in insects is controlled by the Toll and the Imd signaling pathways (De Gregorio et al., 2002). Notably, in honeybees abaecin (GB47318) and hymenoptaecin (GB51223) are controlled by the Imd pathway (Schluns & Crozier, 2007), while others are likely controlled by the Toll pathway (Evans et al., 2006), implying that both pathways are activated in pupae with high SBV levels. In addition, high SBV levels also influenced expression of the Toll pathway genes, including up-regulation of PGRP-SA (GB51741), persephone (GB55007), spatzle (GB52631) and one of the Toll receptors (GB50418), and down-regulation of two other Toll receptors (GB40699and GB43456) (Table 3). We also observed changes in expression of the genes involved in regulation of the melanisation pathway e.g., the simultaneous down-regulation of the prophenoloxidase activating enzyme (PPAE, GB50013), the only honeybee enzyme which proteolytically cleaves prophenoloxidase (Soderhall & Cerenius, 1998) and up-regulation of two putative serpins, the negative regulators of the proteolytic event in the melanisation and signaling pathways (NEC-like proteins, GB48820and GB54611) (Table 3). We propose that these changes in gene expression may result in suppression of the melanisation pathway.

To further explore the possible connection between the replication of DWV and SBV and the expression of the AMPs controlled by the Toll pathway (defensin-1, GB41428), or the Imd pathway (hymenoptaecin, GB51223), and the components of the melanisation pathway (putative serpin, GB48820, and prophenoloxidase activating enzyme, GB50013), we quantified gene expression levels in orally-infected larvae by qRT-PCR (Fig. 1A). While no increase of DWV levels was observed at 4 dpi via the oral route compared to PBS controls, the SBV levels in the virus-infected group were significantly higher than in the control, PBS-exposed insects (Fig. 3A). At 9 days post inoculation, the control insects exposed to the buffer (PBS) or to a UV-inactivated preparation of DWV and SBV (UV-vir) showed similarly low levels of SBV and DWV (Fig. 3B). These results demonstrate that our in vitro manipulations did not activate replication of SBV and DWV that may already have been present at low levels in experimental larvae or pupae.

Figure 3 Oral infection.

The relative levels of SBV and DWV genomic RNAs (A, B), and the AMPs: Imd pathway-controlled hymenoptaecin, GB51223(C, D) and Toll pathway-controlled defensin-1, GB41428(E, F), putative serpin, GB48820(G, H), and prophenoloxidase activating enzyme, PPAE, GB50013(I, J). The number of analyzed larvae for the treatment groups were as follows: for the 4 days post inoculation (d.p.i.) groups n = 6, for the 9 d.p.i groups n = 12. Transcripts were quantified by qRT-PCR. Bars show mean ΔCt values, which were calculated by subtracting Ct values for Rp49 (GB47740) from the Ct values of the target genes, and standard deviation (SD). Bars significantly different at p < 0.01 (using Tukey’s HSD) are indicated using different letters. NS denotes “not significant.”

Figure 4 Pupal injection.

The relative levels of SBV and DWV genomic RNAs (A, B), and the AMPs: Imd pathway-controlled hymenoptaecin, GB51223(C, D) and Toll pathway-controlled defensin-1, GB41428(E, F), putative serpin, GB48820(G, H), and prophenoloxidase activating enzyme, PPAE, GB50013(I, J). The numbers of analyzed pupae for the treatment groups were as follows: for the 2 days post inoculation (d.p.i.) groups n = 6, for the 9 d.p.i groups n = 12. Transcripts were quantified by qRT-PCR. Bars show mean ΔCt values, which were calculated by subtracting Ct values for Rp49 (GB47740) from the Ct values of the target genes, and standard deviation (SD). Bars significantly different at p < 0.01 (using Tukey’s HSD) are indicated using different letters. NS denotes “not significant.”

As before, pupae that developed from larvae fed with infectious virus were stratified according to the observed SBV and DWV levels at 9 dpi (Group “hSBV”—high SBV and low DWV levels, Group “hDWV”—high DWV and low SBV levels, and Group “hSBV/hDWV”—high levels of both tested viruses) and the expression level of honeybee immune genes of interest was quantified (Fig. 3). Both AMPs, hymenoptaecin and defensin-1, were up-regulated in Group “hSBV” insects but remained at control levels in Group “hDWV” individuals (Figs. 3D and 3F). The level of hymenoptaecin increased, but to a lower level in Group “hDWV/hSBV” than Group “hSBV” (Fig. 3D) whereas expression of defensin-1 was similar in these groups (Fig. 3F). It is possible that hymenoptaecin expression may be directly influenced by the level of SBV (which was lower in absolute terms in Group “hSBV/hDWV” than in Group “hSBV”). Alternatively, the elevated levels of DWV in Group “hSBV/hDWV” may suppress Imd pathway activation—which controls expression of hymenoptaecin—but not the Toll pathway-controlled defensin-1. Group “hSBV” and “hSBV/hDWV” samples had elevated expression of the putative serpin and reduced expression of PPAE compared to Group “hDWV” or controls fed PBS or UV-inactivated virus preparation (Figs. 3H and 3J), implying that altered expression of these two melanisation pathway genes could be a result of elevated SBV levels (Fig. 3B). The qRT-PCR analyses were in good agreement with the RNA-Seq data (Table 3).

Injection of honeybee pupae haemolymph with DWV and SBV

High levels of orally-acquired SBV infection has a devastating effect on larval development (Bailey, Gibbs & Woods, 1964). To investigate the influence of the route of virus acquisition on consequent gene expression, we directly inoculated pupae by injection in vitro (Fig. 1B). We observed no pupae with high virus levels in the PBS-injected control group at 2 and 5 dpi, while high levels of DWV were observed in the DWV-injected pupae, and high levels of both SBV and DWV were present in all tested pupae injected with the SBV + DWV virus mixture (Figs. 4A and 4B). As we did not have access to pure SBV preparations due to the presence of DWV in all Warwickshire honeybee colonies (including the SBV-infected used for the virus preparations) and it was not possible to separate these viruses using biophysical methods, no injected pupae with high levels of SBV alone were produced and analyzed.

At 2 dpi there was no significant difference between expression levels of defensin-1 and serpin (Figs. 4E and 4G) whereas the expression of hymenoptaecin were significantly higher in the SBV + DWV-injected pupae compared to DWV-injected pupae (Fig. 4C). In contrast, PPAE levels were higher in DWV-injected pupae than in those receiving both viruses (Fig. 4I). At 5 dpi, PPAE was significantly down-regulated in the SBV + DWV group, while the levels of PPAE in the PBS and DWV groups were not significantly different (Fig. 4J). The same effects of high levels of DWV and SBV on expression of PPAE were observed in the larval feeding experiment (Fig. 3J). Expression levels of hymenoptaecin were significantly different between the pupae injected with PBS, DWV, or SBV + DWV groups at 5 dpi, with the highest levels observed in the SBV + DWV group and lowest in the control (PBS) group (Fig. 4D). In addition, at 5 dpi defensin-1 and serpin (GB48820) were significantly up-regulated in the DWV pupae and SBV + DWV-injected pupae compared to the PBS-injected control. There were no significant differences between the pupae groups with high levels of DWV alone and high levels of both SBV and DWV (Figs. 4F and 4H) at 5 dpi. Notably, high levels of DWV alone in the larval feeding experiment did not alter the expression of defensin-1 and serpin (GB48820) (Figs. 3F and 3H, Group “hDWV”). It is possible that high levels of DWV in the pupae infected by injection may differentially affect the expression of defensin-1 and serpin (GB48820) compared to orally infected larvae. Lourenco et al. (2013) have reported that adult bees exhibit elevated AMP levels following injection. In the absence of bacterial challenge, wounding-associated AMP expression levels decrease within 24 h in bumblebees. Therefore, it is possible that high DWV levels prevent the post-wounding resetting of defensin-1 levels.

Discussion

The Iflaviruses SBV and DWV cause distinctly different disease in A. mellifera with symptoms characteristic to most beekeepers. To better understand the influence of the route of virus transmission on disease development we investigated changes in gene expression resulting from orally administered or injected DWV and SBV. Transcriptome analysis of pupae showed strong up-regulation of the expression of AMPs (defensin-1 and hymenoptaecin) in orally infected larvae with high levels of SBV. Interestingly, high levels of DWV did not up-regulate these AMPs in orally infected larvae (Figs. 3D and 3F). In pupal injection experiments, whilst defensin-1 was equally up-regulated in high DWV and high SBV + DWV groups compared to controls (Fig. 4F), the hymenoptaecin expression showed significantly higher up-regulation in the individual pupae with high levels of both viruses compared with those with high levels of DWV alone (Fig. 4D). This suggests that SBV was a more potent inducer of AMP expression even in injected pupae, where injury alone may have an effect on activation of the signaling pathways and up-regulation of AMPs in the honeybee larvae (Randolt et al., 2008) and young adults (Lourenco et al., 2013).

Expression of AMPs in insects is regulated by the Toll and Imd signaling pathways and induced by recognition of the bacterial or fungal pathogen-associated molecular patterns, such as bacterial peptidoglycan (Lemaitre & Hoffmann, 2007). Our results therefore raise interesting questions including, (i) how replication of SBV, a single-stranded positive-sense RNA virus, activates these signaling pathways and (ii) why DWV, a related Iflavirus with similar genome composition, organization and replication, does not up-regulate AMPs when acquired orally. Although up-regulation of expression of AMPs by RNA viruses has been reported previously (including Drosophila C virus infection of Drosophila (Zhu, Ding & Zhu, 2013) and dengue virus infection of Aedes aegypti (Luplertlop et al., 2011)), it remains unclear how the Toll and Imd signaling pathways are activated by these viruses as they normally respond to the peptigoglycans of Gram-positive and Gram-negative bacteria respectively (Lemaitre & Hoffmann, 2007).

Although the viral proteins encoded by SBV may directly and simultaneously activate the Toll and Imd pathways (which will require further studies) an alternative hypothesis is this results indirectly from SBV-induced pathogenesis. For example, the extensive disruption of the tracheal epithelial lining and pertrophic membranes caused by SBV infection (Mussen & Furgala, 1977) may allow contamination of the haemolymph by bacteria present in the tracheal or intestinal lining. This would result in recognition of the peptidoglycans and consequent Toll and Imd pathway activation. In contrast, DWV infection does not lead to disruption of the gut epithelium (Fievet et al., 2006) and even high levels of DWV, commensurate with symptomatic infection, do not result in AMP up-regulation (Bull et al., 2012; Nazzi et al., 2012; Ryabov et al., 2014). Further molecular studies will be required to discriminate between the direct or indirect activation of Imd and Toll pathways following SBV infection. It should be noted that a simplistic explanation of elevated bacterial levels in SBV-infected pupae does not account for the observations. We quantified the total bacterial load by qRT-PCR using generic primers for bacterial 16S rRNA (Table S1) (Nadkarni et al., 2002) and observed no statistically significant differences between pupae with low and high levels of DWV, SBV or the viruses combined within the same age and developmental stage groups (Fig. S1). However, it is possible that the elevated AMP levels in SBV-infected pupae suppress bacterial expansion so confounding simple quantification of bacterial levels.

Evolution has shaped the virulence and pathogenesis of viruses to facilitate their transmission to new hosts. We speculate that the related Iflaviruses, SBV and DWV, induce different responses in their host that suit their principal or evolutionarily-historical route of transmission. DWV, in the absence of the Varroa mite vectoring, is transmitted vertically via queens and drone semen, and horizontally during trophylaxis (De Miranda & Genersch, 2010; Yue & Genersch, 2005). DWV infection of the honeybee larvae does not halt development and does not cause early death at the larval stage, which suggests that honeybee survival is essential for DWV transmission and that this virus has evolved to minimize negative impact on the host (Fujiyuki et al., 2004; Ryabov et al., 2014). In marked contrast, horizontal oral transmission is considered the principal route for SBV, which causes acute infections at the larval stage leading to death before pupation, with the subsequent spread of SBV likely to involve cannibalization of diseased larvae (Schmickl & Crailsheim, 2001; Woyke, 1977). The observed suppression of the melanisation pathway in the SBV-infected larvae, perhaps a consequence of the combined down-regulation of PPAE and up-regulation of serpin (BeeBase accession number GB48820), may favour SBV transmission as melanisation contributes to virus resistance and could decrease infectivity of SBV particles in the larvae, thereby reducing horizontal transmission (Fig. 5). This may be similar to the Semliki Forest Virus suppression of the phenoloxidase cascade in mosquito (Rodriguez-Andres et al., 2012). In contrast, suppression of melanisation may be detrimental to DWV transmission as it reduces honeybee survival and therefore reduces the opportunities for vertical transmission of this virus (Fig. 5).

Figure 5 Schematic representation of the impacts of SBV and DWV infections on the melanisation pathway, AMP production, host survival and viral transmission.

There is a possibility that up-regulation of AMP expression may prevent bacterial growth and possible degradation of SBV particles in diseased larvae and pupae. Therefore this would increase chances of SBV transmission when diseased larvae and pupae are removed and/or cannibalized as part of the social immune response (Evans & Spivak, 2010) (Fig. 5). Conversely, activation of immune pathways, which result in up-regulation of AMP production is costly (Moret & Schmid-Hempel, 2000) and therefore could negatively impact honeybee survival and ultimately on DWV transmission (Fig. 5).

Conclusions

Our results indicate that evolutionarily-related Iflaviruses SBV and DWV, evoke markedly different transcriptional responses in their honeybee host, including effects on the expression of immune-related genes. We also observed dominance of the SBV-induced transcriptome changes over the DWV-induced. Honeybee larvae with high levels of SBV replication, showed concerted up-regulated expression of antimicrobial peptides (AMPs) and down-regulated expression of the prophenoloxidase-activating enzyme (PPAE) together with up-regulation of the expression of a putative serpin, which could lead to the suppression of the melanisation pathway. The same effect was observed in the individuals with high levels of both SBV and DWV, but high levels of DWV alone did not affect expression of the AMPs and the genes involved in the regulation of melanisation. The effects of high SBV replication levels on expression of these immune genes were unlikely to be the consequences SBV-induced developmental changes, because some of them were observed in honeybees infected with SBV by injection at the pupal stage. It is possible that different impacts of SBV and DWV on the expression of immune-related genes may be an adaptation to horizontal and vertical transmission routes, the principal transmission routes of SBV and DWV respectively. These findings provide the basis for further studies of the contributions of AMPs and melanisation to virus-host interactions and the transmission of insect viruses, including economically important species such as honeybees.

Supplemental Information

Table S1 Oligonucleotides used in this study

Click here for additional data file.

Table S2 Differentially expressed honeybee genes identified in the contrasts of the larval oral inoculation experiment

Click here for additional data file.

Table S3 Over-represented Gene Ontology (GO) terms associated with genes differentially expressed in the honeybees of the larval oral inoculation experiment

Click here for additional data file.

Table S4 Over-represented Gene Ontology (GO) terms associated with differentially expressed genes up-regulated in Contrast 3 (“DWV” versus “Control”) and down-regulated in Contrast 5 (“SBV+DWV” versus “DWV”)

Click here for additional data file.

Table S5 Honeybee immune-related genes differentially expressed in response to SBV and DWV in oral larvae inoculation experiment. Fold change values (log2 transformed) are shown only for the genes DE in the contrasts. Expression of the genes marked with ∗ was quantified by qRT-PCR. DE genes were identified by both DESeq and edgeR analyses, adjusted p < 0.05 and false discovery rate, FDR <0.05

Click here for additional data file.

Figure S1 Quantification of the bacterial load in the infected honeybees. NS, not significant

Click here for additional data file.

We thank Dr. Aronstein for the larval feeding technique.

Additional Information and Declarations

Competing Interests

Author Contributions

Data Availability

The authors declare there are no competing interests.

Eugene V. Ryabov conceived and designed the experiments, performed the experiments, analyzed the data, contributed reagents/materials/analysis tools, wrote the paper, prepared figures and/or tables, reviewed drafts of the paper.

Jessica M. Fannon performed the experiments, contributed reagents/materials/analysis tools, reviewed drafts of the paper.

Jonathan D. Moore analyzed the data, contributed reagents/materials/analysis tools, prepared figures and/or tables, reviewed drafts of the paper.

Graham R. Wood analyzed the data, contributed reagents/materials/analysis tools, wrote the paper, prepared figures and/or tables, reviewed drafts of the paper.

David J. Evans contributed reagents/materials/analysis tools, wrote the paper, reviewed drafts of the paper.

The following information was supplied regarding data availability:

The reported RNA-seq data was deposited into the European Nucleotide Archive under accession number PRJEB6511.

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
