# Peer review of "The Iflaviruses Sacbrood virus and Deformed wing virus evoke different transcriptional responses in the honeybee which may facilitate their horizontal or vertical transmission"

_PeerJ, doi:10.7717/peerj.1591_

## Round 0.1 · original submission · Minor Revisions

This is an interesting study but there is perhaps a bit too much speculation laid out in the abstract and title, especially because your results are based on gene expression only. For example, you do not actually test whether the expression of the AMPs or melanisation pathways alters the ability of these viruses to be transmitted. Please address the concerns of the reviewers, especially reviewer #2 (re: SBV inoculations alone and conclusions therein). I should note that although reviewer #1 did not include much detail in their evaluation, your manuscript was carefully vetted by me, the handling editor.

Reviewer 1 ·

Basic reporting

35: deformed wing virus -> at least ‘deformed’ is written with a capital first letter
37: Please change the italic reference.
40-41: Although DWV has a limited genetic diversity, there a some hotspots like Lp. This gene is very diverse, even in one apiary (Ravoet 2014).
49-50: It might be relevant to discuss the severe impact of the Asian SBV serotypes on the development of Apis cerana.
76: virus preparations stored -> virus preparations were stored
150, 344: Mention the databank for this accession number, like GenBank.

Experimental design

69: It is unclear to me if the virus preparations were checked for viral contaminants since multiple viruses can be present in one honeybee?

Validity of the findings

No Comments

Additional comments

Indentations below a title are not necessary.

Reviewer 2 ·

Basic reporting

Overall the article is well written, and frames the study succinctly in terms of honeybee decline. However, the introduction fails to properly introduce the later discussion of viral evolutionary strategy which forms the link between the scale of this study and its relevance to honeybee declines. This will be particularly important if recommendations below on expanding the discussion of adaptive virulence are taken on board.
Figures are clear, informative, and do an excellent job of expressing all necessary information.

Experimental design

The experimental concept is certainly novel enough to warrant publication in PeerJ, hypotheses are clearly identified and there is no doubt about the clarity of the proximate research questions. However there are problems with design, implementation and reporting of the experiment.
While the treatments are suitably controlled – I am particularly pleased to see the use of both a saline treatment and an inactivated viral treatment - there is a conspicuous absence of an SBV-only pupal injection treatment (Lines 89-95, 263-291, Fig 4). This omission needs to be rectified, or as a minimum adequately justified; it currently leaves this aspect of the study incomplete and not is addressed anywhere in the text. This missing treatment casts doubt on some of the claims made in the discussion (further comments below).
Additionally, there appears to be nowhere in the manuscript or supplementary information details as to how many larvae and pupae were treated. This absence of sample size is a basic oversight which should have been addressed before submission. Alongside no reporting of degrees of freedom for the t-tests undertaken in figures 3 and 4, this is not acceptable for publication.
The suite of molecular methods and subsequent bioinformatics used to characterise the responses to the treatments are thorough, well analysed, and presented clearly.
However, care should be taken that all acronyms are stated in unabbreviated form first, even if common across the discipline:
Line 119 – ‘NGS’ never stated as next-generation sequencing

Validity of the findings

Given the unaccounted for omission of the SBV-only injection treatment, a questionable assertion is made in discussion: Lines 295 – 297
“The pupal injection experiment further confirmed that hymenoptaecin and defensin-1 are up-regulated in the insects with high SBV levels.”,
summarise this. I don’t believe this claim can be made without an SBV-only injection treatment, particularly as there are disparities between the oral and injection infection results for DWV (Lines 286-288). Much of the remaining discussion relies on this contentious claim, and ultimately the conclusions drawn are not convincingly supported by the data as a result.
Setting the above problem aside, discussion of the alternative mechanisms for SBV causing up-regulation are fairly assessed, with clear steps taken (Fig. S1) to address competing explanations. The nuanced discussion in this paragraph (Lines 309 – 327) sets a clear mandate for further study on how to detect direct vs indirect effects mediated by SBV pathogenesis.
The framing of the observed differences in transcriptome response to DWV or SBV infection in an adaptive virulence context (Lines 328-352) is commendable, however I would recommend this aspect of the discussion to be refined and extended. It is not clear why, when horizontal transmission has been shown as possible in both viruses, they have diverged in strategy. There is no mention of SBV as a potential obligate killer, which is what Fig. 5 portrays. More considered reasoning and acknowledgement should be shown of the positive feedback between high pathogenesis leading to transmission principally following host death, and the establishment of this main mode of transmission reducing remaining selective constraints on pathogen virulence.
As a recommendation, this aspect of the discussion would substantially improve the paper if it also framed the observed dominance on the transcriptome of SBV over DWV in the context of co-infection. It is clear the authors are suggesting opposing transmission strategies between SBV and DWV, and considering co-infection was administered in the study it is again conspicuously absent from the discussion. This point in particular highlights the study’s potential relevance to the originally introduced problems of honeybee decline in the face of virulence pathogens.

Additional comments

Overall I am keen to see this study live up its potential, and can readily recognise the contribution of the manuscript to the wider literature, should the recommendations be taken on board. The main concern is the conspicuous absence of a SBV-only injection treatment. Addressing that missing aspect, in addition to the other comments on sample size and discussion emphasis, will make for a very compelling article. However in its current state, I do not consider this submission of publishable quality.

---

## Round 0.2 · accepted · Accept

The manuscript has been much improved and I agree with Reviewer #2 that the interpretation of the data is now adequate. Happy holidays.

Reviewer 2 ·

Basic reporting

No comments

Experimental design

No comments

Validity of the findings

No comments

Additional comments

I'm happy with the changes made, outlined in the rebuttal letter, leading me to recommend this article for publication. Given the impossibility of an SBV-only treatment, I agree that the changes made in reporting, and additional sentence highlighting coinfective dominance of SBV, are adequate improvement. The further qualification of your infection dynamic interpretation as highly speculative is adequate, and indirectly emphasises the (rightful) possibility of alternative explanations. In my view, exploration of them is not necessary for this manuscript's publication in PeerJ.

Whilst not explicitly stated, it is now clear to readers from the information in the study that SBV infection in A. mellifera is almost universally additional to DWV circulation within a colony - hence the lack of SBV-alone injection observations. This exclusively coinfective ecology of SBV could, speculatively, be the evolutionary driver leading to its observed dominance over DWV. This line of reasoning makes the manuscript of increased interest, but admittedly this study is not the place for its discussion.

I would also like to apologise for my oversight in not properly checking figure legends for sample sizes, when claiming they were not present in the manuscript.

I look forward to future work on this following the manuscript's publication. I believe there is substantial potential in this DWV-SBV system as a fruitful line of both A. mellifera ecology and pathogen evolution research.

---

## Author Rebuttal · Round 0.2

11 December 2015

Dear Dr Newton

Thank you for considering our submission for publication and for the suggestions. We are pleased that PeerJ found our study interesting. Below are detailed answers to the questions. We have also submitted a revised version of the manuscript with tracked changes.

Sincerely

Eugene Ryabov

*From: PeerJ <peer.review@peerj.com>*
*Sent: 09 November 2015 17:20*
*To: Ryabov, Eugene*
*Subject: Decision on your PeerJ submission: "Evolutionarily related Sacbrood virus and Deformed wing virus evoke different transcriptional responses in the honeybee which may facilitate horizontal or vertical transmission of these viruses" (#2014:10:2936:0:0:RE...)*

*Thank you for your submission to PeerJ. I am writing to inform you that in my opinion as the Academic Editor for your article, your manuscript "Evolutionarily related Sacbrood virus and Deformed wing virus evoke different transcriptional responses in the honeybee which may facilitate horizontal or vertical transmission of these viruses" (#2014:10:2936:0:0:REVIEW) requires some minor revisions before we could accept it for publication.*

*The comments supplied by the reviewers on this revision are pasted below. My comments are as follows:*

***Editor's comments***

*This is an interesting study but there is perhaps a bit too much speculation laid out in the abstract and title, especially because your results are based on gene expression only. For example, you do not actually test whether the expression of the AMPs or melanisation pathways alters the ability of these viruses to be transmitted. Please address the concerns of the reviewers, especially reviewer #2 (re: SBV inoculations alone and conclusions therein). I should note that although reviewer #1 did not include much detail in their evaluation, your manuscript was carefully vetted by me, the handling editor.*

We have revised the "Abstract" by removing parts which were speculative, leaving only an overview of the factual findings and have included the following sentence: *"We propose that the difference in expression of the honeybee immune genes induced by SBV and DWV may be an evolutionary adaptation to the different predominant transmission routes used by these viruses."*

We slightly modified the title, the new version *"The Iflaviruses Sacbrood virus and Deformed wing virus evoke different transcriptional responses in the honeybee which*

*may facilitate their horizontal or vertical transmission*". We believe that the modified title accurately reflects the paper content. The first part of the title "*The Iflaviruses Sacbrood virus and Deformed wing virus evoke different transcriptional responses in the honeybee …*" states that this is a transcriptome analysis study, while the second part of the title "…*which __may facilitate__ their horizontal or vertical transmission*" clearly indicates that there is only a possibility of the connection between the differences of immune gene expression and the transmission routes.

As requested by Reviewer 2, we specifically explained the lack of a high SBV alone group in the pupae injection experiments. ("…As we did not have access to pure SBV preparations (due to the presence of DWV in all Warwickshire honeybee colonies including the SBV-infected used for the virus preparations) no injected pupae with high levels of SBV alone were produced and analyzed.)

*If you are willing to undertake these changes, please submit your revised manuscript (with any rebuttal information\*) to the journal within 45 days.*
*\* Resubmission checklist:*
*When resubmitting, in addition to any revised files (e.g. a clean manuscript version, figures, tables, which you will add to the "Primary Files" upload section), please also provide the following two items:*

*A rebuttal Letter: A single document where you address all the Editor and reviewers' suggestions or requirements, point-by-point.*
*A 'Tracked Changes' version of your manuscript: A document that shows the tracking of the revisions made to the manuscript. You can also choose to simply highlight or mark in bold the changes if you prefer.*
*Accepted formats for the rebuttal letter and tracked changes document are: DOCX (preferred), DOC, or PDF.*
*PeerJ does not offer copyediting, so please ensure that your revision is free from errors and that the English language meets our standards: uses clear and unambiguous text, is grammatically correct, and conforms to professional standards of courtesy and expression.*
*Irene Newton*
*Academic Editor for PeerJ*
*Reviewer Comments*

### *Reviewer 1 (Anonymous)*

*Basic reporting*

*35: deformed wing virus -> at least 'deformed' is written with a capital first letter*

Corrected as advised Deformed wing virus (DWV)

*37: Please change the italic reference.*

Corrected as advised

*40-41: Although DWV has a limited genetic diversity, there a some hotspots like Lp. This gene is very diverse, even in one apiary (Ravoet 2014).*

Changed "limited genetic diversity" to "significantly reduced genetic diversity", to be more precise. Significant reduction of DWV genetic

diversity in Varroa-infested honeybee colonies was specifically reported in Martin at al 2012 and Ryabov et al 2014. Although there were some "hot spots" of higher diversity in Varroa-associated DWV populations, including Leader Protein-coding region (Lp), we did not specify this in this paragraph.

*49-50: It might be relevant to discuss the severe impact of the Asian SBV serotypes on the development of Apis cerana.*

We have added to the sentence in question a statement on SBV infection in Asian bees "…*Sacbrood virus, SBV, a related member of the Iflaviridae) a predominant viral pathogen in the Asian honeybee Apis cerana,* …: and the reference "Ai H, Yan X, Han R. 2012. Occurrence and prevalence of seven bee viruses in Apis mellifera and Apis cerana apiaries in China. Journal of Invertebrate Pathology 109:160–164 DOI:10.1016/j.jip.2011.10.006" was also included.

*76: virus preparations stored -> virus preparations were stored*

Corrected as suggested.

*150, 344: Mention the databank for this accession number, like GenBank. Experimental design*

Line 150 – *"GenBank accession numbers NC_004830 and AF092924 respectively"*)
Line 344 – "..serpin (BeeBase accession number GB48820)..

*69: It is unclear to me if the virus preparations were checked for viral contaminants since multiple viruses can be present in one honeybee?*

Apart from DWV and SBV, we detected no other honeybee viruses in the preparations used in this study. A sentence has been added "No other known honeybee viruses were detected in the preparations"

*Validity of the findings*

*No Comments*
*Comments for the author*

*Indentations below a title are not necessary.*

### *Reviewer 2 (Anonymous)*

*Basic reporting*

*Overall the article is well written, and frames the study succinctly in terms of honeybee decline. However, the introduction fails to properly introduce the later discussion of viral evolutionary strategy which forms the link between the scale of this study and its relevance to honeybee declines. This will be particularly important if recommendations below on expanding the discussion of adaptive virulence are taken on board.*

*Figures are clear, informative, and do an excellent job of expressing all necessary information.*

We are very pleased that Reviewer 2 found the results of this study and the discussion relevant to honeybee decline. We are also pleased that the Reviewer appreciated the effort we invested in preparing Figures and Tables to illustrate the findings and discussion points.

*Experimental design*

*The experimental concept is certainly novel enough to warrant publication in PeerJ, hypotheses are clearly identified and there is no doubt about the clarity of the proximate research questions. However there are problems with design, implementation and reporting of the experiment.*
*While the treatments are suitably controlled – I am particularly pleased to see the use of both a saline treatment and an inactivated viral treatment - there is a conspicuous absence of an SBV-only pupal injection treatment (Lines 89-95, 263-291, Fig 4). This omission needs to be rectified, or as a minimum adequately justified; it currently leaves this aspect of the study incomplete and not is addressed anywhere in the text. This missing treatment casts doubt on some of the claims made in the discussion (further comments below).*

In the pupae injection experiments we were using virus preparations, pure DWV (DWV) and mixed SBV and DWV (SBV+DWV), but not SBV alone (as described in Materials and Methods). Injection of the honeybee pupae with the DWV alone preparation resulted in development of high levels of DWV, whilst injection of the mixed SBV+DWV always resulted in replication of both SBV and DWV to high levels. We were unable to obtain virus preparations of SBV alone, because DWV was present in all UK honeybee colonies in the UK, including the SBV-infected used to isolate virus preparations. Both SBV and DWV are evolutionary related viruses with very similar sizes and buoyant densities of their virus particles, making it impossible to separate these viruses using biophysical methods (e.g by centrifugation). It is also diffcult to produce DWV-free SBV preparation because (i) that the random apiary survey in the UK showed that DWV was present in at least 95% of tested colonies, (ii) that our previous published studies have never failed to detect DWV in all pupae – even those no exposed to Varroa, (iii) there is no cell culture methods available to propagate these viruses.

As requested by the reviewer, we have now explained the lack of pupae with high levels of SBV alone in the injection experiments ( page 12, "Injection of honeybee pupae haemolymph with DWV and SBV", end of the first paragraph, underscored : "*We observed no pupae with high virus levels in the PBS-injected control group at 2 and 5 dpi, while high levels of DWV were observed in the DWV-injected pupae, and high levels of both SBV and DWV were present in all tested pupae injected with the SBV+DWV virus mixture (Fig. 4A, B) with the multiplicity of infection used in this study. As we did not have access to pure SBV preparations due to the presence of DWV in all Warwickshire honeybee colonies (including the SBV-infected used for the virus preparations) and it was not possible to separate these viruses using biophysical methods, no injected pupae with high levels of SBV alone were produced and analyzed.*"

*Additionally, there appears to be nowhere in the manuscript or supplementary information details as to how many larvae and pupae were treated. This absence of sample size is a basic oversight which should have been addressed before submission. Alongside no reporting of degrees of*

*freedom for the t-tests undertaken in figures 3 and 4, this is not acceptable
for publication.*

We included information about the size of the samples in the legends to Figures 3 and 4. Figure 3: "… *The numbers of analyzed larvae for the treatment groups were as follows: for the 4 days post inoculation (d.p.i.) groups n=6, for the 9 d.p.i groups n=12. ...*" ; Figure 4: " ... The numbers of analyzed pupae for the treatment groups were as follows: for the 2 days post inoculation (d.p.i.) groups n=6, for the 9 d.p.i groups n=12. ...". We also provided more information of statistical analysis of the qPCR results: in the "Materials and Methods" / end of the "Gene expression analysis" section (page 5) "… *Tukey's Honest Significant Difference test (Tukey's HSD) was used to determine significantly different virus and gene expression levels. ...*". In the Legends to Figures 3 and 4 we specified that "… *Bars significantly different at p<0.01 (using Tukey's HSD) are indicated using different letters. …*"

*The suite of molecular methods and subsequent bioinformatics used to characterise the responses to the treatments are thorough, well analysed, and presented clearly. However, care should be taken that all acronyms are stated in unabbreviated form first, even if common across the discipline: Line 119 – 'NGS' never stated as next-generation sequencing*

Un-abbreviated "Next generation sequencing (NGS)" is included in line 119.

*Validity of the findings*

*Given the unaccounted for omission of the SBV-only injection treatment, a questionable assertion is made in discussion: Lines 295 – 297 "The pupal injection experiment further confirmed that hymenoptaecin and defensin-1 are up-regulated in the insects with high SBV levels.", summarise this. I don't believe this claim can be made without an SBV-only injection treatment, particularly as there are disparities between the oral and injection infection results for DWV (Lines 286-288). Much of the remaining discussion relies on this contentious claim, and ultimately the conclusions drawn are not convincingly supported by the data as a result.*

We revised the part of the Discussion related to injection experiments. In particular, we omitted the sentence (Lines 295 – 297) *"The pupal injection experiment further confirmed that hymenoptaecin and defensin-1 are up-regulated in the insects with high SBV levels …".* We also pointed out that the "high SBV-high DWV" group clearly showed that, in hymenoptaecin is significantly higher expressed in this group compared to the "high DWV alone" by including the following part: "*In the pupal injection experiment, whilst defensin-1 was equally up-regulated in the high DWV and the high SBV-DWV groups compared to the PBS injected pupae (Fig 4 F), the hymenoptaecin expression showed significantly higher up-regulation in the individual pupae with high levels of both SBV and DWV compared with those with high levels of DWV alone (Fig. 4 D). This suggests that SBV was a more potent inducer of AMP expression even in the injected pupae, where injury alone may have an effect on activation of the signaling pathways and up-regulation of AMPs (Randolt et al, 2008; Lourenco et al, 2013).*"

*Setting the above problem aside, discussion of the alternative mechanisms for SBV causing up-regulation are fairly assessed, with clear steps taken (Fig. S1) to address competing explanations. The nuanced discussion in this paragraph (Lines 309 – 327) sets a clear mandate for further study on how to detect direct vs indirect effects mediated by SBV pathogenesis.*

We are very pleased that Reviewer 2 acknowledged that the results of the effect of SBV on the honeybee immune gene expression presented in our study warrant further research attention.

> *The framing of the observed differences in transcriptome response to DWV or SBV infection in an adaptive virulence context (Lines 328-352) is commendable, however I would recommend this aspect of the discussion to be refined and extended. It is not clear why, when horizontal transmission has been shown as possible in both viruses, they have diverged in strategy. There is no mention of SBV as a potential obligate killer, which is what Fig. 5 portrays. More considered reasoning and acknowledgement should be shown of the positive feedback between high pathogenesis leading to transmission principally following host death, and the establishment of this main mode of transmission reducing remaining selective constraints on pathogen virulence.*

We clearly stated in this section of the paper (from lines 328)"… *We speculate that the related Iflaviruses, SBV and DWV*, … " provided a possible explanation of the observed differences in the gene expression in response to SBV and DWV. We believe that this suggestions set up areas for future studies.

It was already specifically mentioned in this section that SBV is a highly pathogenic virus, which kills honeybee larvae (line 340) "*SBV, which causes acute infections at the larval stage leading to death before pupation…*,

> *As a recommendation, this aspect of the discussion would substantially improve the paper if it also framed the observed dominance on the transcriptome of SBV over DWV in the context of co-infection. It is clear the authors are suggesting opposing transmission strategies between SBV and DWV, and considering co-infection was administered in the study it is again conspicuously absent from the discussion. This point in particular highlights the study's potential relevance to the originally introduced problems of honeybee decline in the face of virulence pathogens.*

Thank you for suggesting further highlighting this important point. We mentioned in the results that the effects of SBV on the honeybee gene expression "override" the effect of DWV, and that DWV and SBV have opposite effects on the expression of some honeybee genes (see lines 197-200 and Supplementary Table S4. "Over-represented Gene Ontology (GO) terms associated with differentially expressed genes up-regulated in Contrast 3 ("DWV" versus "Control") and down-regulated in Contrast 5 ("SBV+DWV" versus "DWV")."

In the revised version we made this point clearer, by specifically adding the statement after the first sentence of the "Conclusions" section (line 364) "*We also observed dominance of SBV-induced transcriptome changes over the DWV-induced.* "

> *Comments for the author*

> *Overall I am keen to see this study live up its potential, and can readily recognise the contribution of the manuscript to the wider literature, should the recommendations be taken on board. The main concern is the conspicuous absence of a SBV-only injection treatment. Addressing that missing aspect, in addition to the other comments on sample size and discussion emphasis, will make for a very compelling article. However in its current state, I do not consider this submission of publishable quality.*